# Predictive Modelling of Sports Facility Use: A Model of Aquatic Centre Attendance

Antoine Barbier *⬛, Barbara Evrard and Nadine Dermit-Richard

Faculty of Sport Sciences, University of Rouen Normandy, CETAPS UR3832, 76821 Mont-Saint-Aignan, France
* Correspondence: barbier.antoine.pro@gmail.com

**Abstract:** The level of attendance appears to be the central indicator for analysing the performance of public sports facilities. However, most of the studies focus on customer satisfaction and loyalty and have been carried out in Australia and the United Kingdom. The aim of this article was, therefore, to identify potential explanatory variables that could explain attendance at leisure sports facilities in Europe based on the literature (a). Then, we aimed to identify the variables that explained attendance based on a study of aquatic centres (b) to propose an exploratory predictive model (c). The sample was composed of data from 28 aquatic facilities over 5 years, and we examined 41 variables from the literature. A predictive model of attendance was created using backward regression. The proposed formula had a predictive power of 79.13% of the observed attendance in our sample of aquatic centres. These results suggest that it is possible to determine attendance at an aquatic facility with only four variables and that the study of leisure facilities in Europe implies adapting the variables to be considered. This is also the first model to investigate leisure sports facilities in Europe.

**Keywords:** attendance; aquatic centres; predictive model; sports facilities

## 1. Introduction

In the twenty-first century, there has been a growing demand for a reduction in public expenditure, particularly in the non-governmental sectors. Simultaneously, consumer demand for sport and leisure has increased, becoming more complex and diverse. Moreover, sedentary lifestyle has become a public health problem that has influenced changes in the public policy. Leisure sports facilities are a part of the solution to adopt a physically active lifestyle, insofar as these facilities are accessible to the largest possible number of people.

The state and local authorities are the main owners of these facilities in Europe [1–3], the United States [4], Australia [5], and Japan [6]. In addition, in France, public authorities own 87% of the 318,000 installations identified [7–10].

These leisure facilities are continuously diversifying as interest in health, wellness, and high-quality recreation intensifies [11–14]. They are more versatile but are also more expensive. The costs of building and operating public infrastructure are mainly supported by public authorities. The question regarding the relationship between the cost of these facilities and the services provided to users is even more acute at the time of public spending cuts. In a context of increasing budgetary constraints, the objective of these operating authorities is to optimise the relationship between attendance and operating costs [15,16]. In the business world, the performance of an operation is generally measured according to profit, and the evaluation of the performance of facilities intended to perform public service missions must also be understood in terms of social utility and optimisation of the services provided to users [17,18]. For this reason, the performance of a public sports facility is generally assessed in the literature based on its level of attendance [2,17,19,20].

It is, therefore, essential to understand the determinants of facility attendance to model it and make the most robust forecasts possible from an exploratory perspective. The objective is to be able to determine its size within the framework of a reflection on the

implementation of a new facility [19]. However, the choice of capacity determines both the cost of building the facility and that of operating it over its lifetime [18]. In the case of professional sports facilities, the issue of explaining and predicting attendance has been widely studied [21,22]. However, limited work has been conducted to date on leisure sports facilities [19].

In the current context of budgetary constraints imposed on public actors who finance these facilities, it seems interesting to provide decision-making tools to better evaluate the potential attendance at a leisure sports facility, considering its location and the services it can provide to the population. This is the purpose of this contribution. To this end, we sought to build a predictive model of aquatic centre attendance by considering the characteristics of the facilities and the specificities of the area in which they are located. These facilities, which combine indoor and outdoor spaces, are particularly well-suited to hosting competitive, recreational, and wellness activities. There are approximately 3200 aquatic facilities in France, of which 2850 are managed by non-market organisations (public authorities or associations) and 350 are managed by private companies [9]. Of EUR 20 billion of public spending on the sports sector as a whole [10], aquatic facilities represent the largest expenditure, with an average annual operating cost of EUR 1.2 million per facility [23]. These are also the facilities that are logically the most affected by current inflation, which generates additional costs estimated at an average of EUR 350 K per aquatic facility [10]. In response to these increasing operating costs, several public and private operators have decided to close their facilities, usually temporarily. These characteristics make them particularly interesting to study and are the subject of a large number of publications on leisure sports facilities [24]. However, the methodology proposed in this case is applicable to any category of leisure sports facilities in any country.

To that purpose, we first conducted a literature review on this issue and then devised the methodology to build our model. We present the results and conclude with an analysis and discussion of these results.

## 2. Literature Review

In a context of policy change that aims to rationalise public spending, the rationale for subsidising sports facilities has been widely investigated, leading to questions about the performance of their operations and financial sustainability [15,18].

Indeed, the performance of the operation of a sports facility can be viewed from different perspectives [18]. Economic performance is the most commonly used indicator in the private sector. Performance can also be considered from the perspective of "Efficiency", which focuses on achieving objectives and targets at minimum cost and considers the best possible relationships between inputs and outputs. "Effectiveness" is solely concerned with the achievement of output targets. It is an important performance aspect in public sector leisure services, as they are concerned with social objectives that are largely non-financial in nature.

In an analysis of the performance of aquatic facilities in England over the period of 2005–2016 along these three dimensions, Ramchandani, Shibli, and Kung [18] (p. 12) concluded as follows: "Our results also show that the improvement in the overall financial efficiency of public sport facilities has not been achieved as a result of cost savings, but rather by focusing on revenue generation" (which depends on attendance). Iversen [17] adopted the same position, considering that the objective of publicly funded facilities is to provide a place for physical activity in order to meet societal objectives (sports health and social demand). Consequently, the objective of public policy is that the facilities should benefit as many people as possible. Therefore, operational performance should aim to maximise attendance at the facility [17] (competitiveness for maximising utilisation).

Ramchandani, Shibli, and Kung [18] also identified another dimension of performance that is increasingly common, customer/user satisfaction, which is indicated by the comparison between consumers' initial expectation and their final opinion on the service rendered [24]. Several studies on this topic have used a managerial approach

to identify the impact of operating methods on quality, overall satisfaction, and loyalty. The final objective is to determine the operating methods that influence attendance at leisure sports facilities [3,25,26] and aquatic facilities in particular [24,27–29]. Recent studies have shown the impact of the quality of the processes linked to the operating methods on the one hand, and the quality of the results on the other, on the creation of value, overall satisfaction, and the conditions for loyalty. They have focused on fitness centres in Malaysia [30] and Spain [31,32] and on aquatic facilities in Australia [33,34] and Hong Kong [24]. All these studies used indicators related to the operational phase within the structure. Only Lau et al. [24] analysed customer satisfaction based on a grid co-constructed with users, including architectural elements (size, pool tank, pool wall, and pool deck) in addition to building service aspect factors (air, temperature, lighting, water, and acoustics). However, they showed that the latter elements were largely predominant in user satisfaction.

The question of analysing the determinants of attendance has been the subject of numerous publications in the case of professional sports facilities [21,22], allowing us to identify four categories of variables that have an impact on attendance at these facilities. The first category comprises the criteria related to the appeal of the proposed activity. The second category includes economic variables, such as ticket prices. The third category groups together the socio-demographic data of the population, such as market size, age, sex, ethnic origin, profession, education, and location of the event. Moreover, the fourth category concerns public preferences and refers to consumption habits, which include time of the event, ease of access, amenities, weather, and the overall quality of the stadium. These studies statistically analyse the link between variables that could potentially explain attendance and attendance at these facilities.

The only study that used this methodology in the context of leisure sports facilities concerned Australian aquatic centres [19]. The authors constructed an explanatory model of aquatic centre attendance (number of entries per year) using four variables: the population of the catchment area within a radius of 5 km around the facility, communication expenses, the number of activities offered per week, and the entrance fee. The model explained 54.6% of attendance. However, they did not include socio-demographic variables in their work.

Moreover, Schreyer and Ansari [21] pointed out the lack of studies on the determinants of attendance at athletics and swimming competitions, and the scarcity of studies dealing with this issue in recreational sports facilities.

Our study is based on the work of Howat, Murray, and Crilley [19] on Australian public aquatic centres, in which the level of attendance appeared to be the central indicator for analysing the performance of these public facilities. We supplemented their study with international literature studies on attendance at professional sports facilities in order to identify and test new variables. Our objective was to complement the work on the analysis of aquatic facility attendance using a quantitative approach adapted to the European context.

Two main hypotheses guided our work. Our first hypothesis was that the variables presented below (see Table 1), according to the four categories identified in the literature, explain attendance at aquatic facilities in a European context.

The second hypothesis was that from the variables correlated with attendance, it is possible to determine a predictive model adapted to the European context.

The aim of this article was, therefore, to identify potential explanatory variables that could explain attendance at leisure sports facilities in Europe based on the literature (a). Then, we aimed to identify the variables that explained attendance based on a study of aquatic centres (b) to propose an exploratory predictive model (c). More precisely, we aimed to build a predictive model for French aquatic centre attendance by considering the characteristics of the facilities and the specificities of the area where they are located.

**Table 1.** Potential explanatory variables of attendance based on the literature review.

| | Variables | Previous Studies |
|---|---|---|
| 1. | Category 1: Criteria related to the appeal of the proposed activity | |
| 1.1. 1.2. | Number of hours when the centre is open to the public Number of hours when the various activities are offered | Howat, Murray, and Crilley 2005 [19]; Howat and Assaker 2013 [29]; Anderson, Ramos, and Middlestadt 2014 [28]; Afthinos, Theodorakis, and Howat 2018 [27] |
| 1.3. | Temperature of pool water | Lau et al. 2021 [24]; Howat and Assaker 2016 [33]; Liu, Taylor, and Shibli 2008 [35] |
| 1.4. | Area available for each of the activities (wellness and fitness) | Lau et al. 2021 [24] |
| 1.5. | Number of swimming lanes | Lau et al. 2021 [24] |
| 1.6. | Communication budget | Howat, Murray, and Crilley 2005 [19] |
| 2. | Category 2: Economic variables | |
| 2.1. | Normal entrance fee for adults | Howat and Crilley 2007 [36] |
| 2.2. | Number of direct competitors in the catchment area | Howat and Crilley 2007 [36] |
| 3. | Category 3: Variables characterizing the demographics of the catchment area of each centre | |
| 3.1. | Variables related to overall population (total and gender) | Howat, Murray, and Crilley 2005 [19]; Afthinos, Theodorakis, and Howat 2018 [27]; García-Pascual et al., 2021 [32] |
| 3.2. | Variables related to population age groups | Liu, Taylor, and Shibli 2008 [35]; 2009 [14]; Afthinos, Theodorakis, and Howat 2018 [27]; García-Pascual et al., 2021 [32] |
| 3.3. | Population based on profession and socio-professional category | Liu, Taylor, and Shibli 2008 [35]; Afthinos, Theodorakis, and Howat 2018 [27] |
| 3.4. | Variables related to the economic dynamics of the territory | Scelles et al. 2013 [37] |
| 4. | Category 4: Variables related to "public preferences" | |
| 4.1. | Number of public transport means | Callède 2007 [38] |
| 4.2. | Meteorological variables (sunshine and precipitation) | Howat, Crilley, and Mcgrath 2008 [34]; Anderson, Ramos, and Middlestadt 2014 [28] |

## 3. Methods

The combination of the different works in our literature review allowed us to identify variables that may explain attendance and that can be budgeted for in a prospective model. Research on professional sports facilities [22] has identified broad categories for classifying variables and better distinguishing indicators (see Table 1).

From our study, we excluded the variables identified in the literature related to service quality [33] that could not be budgeted for (e.g., quality of reception, cleanliness of premises, etc.).

### 3.1. Data Collection

The analysis of the performance of public leisure sports facilities was confronted with a lack of access to data. Public accounting only produces data on costs and public subsidies but does not collect quantitative data such as attendance. Research in England and Australia on these public facilities has been based on data collection organised as part of publicly funded research programmes conducted by academics through National Benchmarking Service (NBS) [35] and Centre for Environmental and Recreation Management

(CERM) [19], respectively. Furthermore, in the French case, the only way to access these data is to work on the annual public service delegation (PSD) reports produced by private companies that operate an aquatic establishment under delegated management. These reports are public. However, the information provided is not standardised and depends on the operators concerned. Therefore, we worked on data in reports published by the largest private operator in France. This also avoided methodological biases linked to each operator's aquatic centre management policy. We thus had access to homogeneous data on a sample of 28 facilities covering a five-year period: 2015 to 2019.

The study was conducted in three steps:

- First, we determined the variable to be explained: attendance at the facility. Based on the definitions given in the literature [19,21], we operationalised this variable using the number of individuals who paid an entrance fee to the facility, who we then considered customers. We determined attendance by taking the average of the attendance observed over the 5 years of the study, weighted by the number of annual opening days (source: annual reports for the PSD).
- Second, we defined 41 potentially explanatory variables of aquatic centre attendance based on our literature review (see Table 1) and the data available for our research. The potential explanatory variables selected are listed in Table 2.

**Table 2.** Potential explanatory variables of attendance collected in the study.

| | Variables | Sources |
|---|---|---|
| 1. | Category 1: Criteria related to the appeal of the proposed activity | |
| 1.1. | Number of hours when the centre was open to the public (2015–2019) | Facility operators |
| 1.2. | Number of hours when the various activities were offered (2015–2019) | Facility operators |
| 1.3. | Temperature of pool water (2015–2019) | Facility operators |
| 1.4. | Area available for each of the activities (wellness and fitness) 1.4.1. Total area of the centre in $m^2$ 1.4.2. Total area of all the activity spaces in $m^2$ 1.4.3. Total area of all the spaces for children's activities in $m^2$ 1.4.4. Total area of all the wellness spaces in $m^2$ 1.4.5. Total area of all the fitness spaces in $m^2$ 1.4.6. Total area of all the spaces for group lessons in $m^2$ | Technical data sheet |
| 1.5. | 1.5.1. Number of swimming lanes 1.5.2. Total area of all pools in $m^2$ | Technical data sheet |
| 1.6. | Communication budget (2015–2019) | Facility operators |
| 2. | Category 2: Economic variables | |
| 2.1. | Normal entrance fee for adults (2015–2019) | Annual reports for the PSD |
| 2.2. | Number of direct competitors in the catchment area | Inventory of sports facilities |
| 3. | Category 3: Variables characterizing the demographics of the catchment area of each centre | |
| 3.1. | Variables related to the overall population 3.1.1 Number of inhabitants in the municipality 3.1.2. Number of households in the municipality 3.1.3. Male population in the municipality 3.1.4. Female population in the municipality 3.1.5. Number of inhabitants in the municipality of 15 years old or older | French Institute of statistics and economic studies (INSEE)—Census 2013 |

**Table 2.** *Cont.*

| | Variables | Sources |
|---|---|---|
| 3.2. | Variables related to population age groups<br>3.2.1. Number of inhabitants in the municipality between 0 and 14 years<br>3.2.2. Between 15 and 29 years<br>3.2.3. Between 30 and 44 years<br>3.2.4. Between 45 and 59 years<br>3.2.5. Between 60 and 74 years<br>3.2.6. Between 75 and 89 years<br>3.2.7. 90 years old or more | French Institute of statistics and economic studies (INSEE)—Census 2013 |
| 3.3. | Population based on profession and socio-professional category<br>3.3.1. Number of inhabitants in the municipality of 15 years old or older self-employed in the agricultural sector<br>3.3.2. Self-employed<br>3.3.3. Doing intellectual or managerial work<br>3.3.4. Mid-level jobs: mid-way between management posts and agents, workers, or employees<br>3.3.5. Salaried intellectual work<br>3.3.6. Salaried manual work<br>3.3.7. Without employment and not receiving unemployment compensation but who were once employed<br>3.3.8. Out of work in 2013 | French Institute of statistics and economic studies (INSEE)—Census 2013 |
| 3.4. | Variables related to the economic dynamics of the territory<br>3.4.1. Value that divides into two equal parts the standard of living of the households in the municipality<br>3.4.2. Number of persons employed in the municipality<br>3.4.3. Number of persons receiving unemployment compensation<br>3.4.4. Number of firms operating in the municipality on 31 December 2014 | French Institute of statistics and economic studies (INSEE)—Census 2013 |
| 4. | Category 4: Variables related to "public preferences" | |
| 4.1. | Number of public transport means (with stops within 300 m of the centre) | Municipalities and Google Maps |
| 4.2. | Meteorological variables (sunshine and precipitation)<br>4.2.1. Number of sunny days in year n (2015–2019)<br>4.2.2. Number of rainy days in year n for each centre (2015–2019) | Météo France |

The annual data from 2015, 2016, 2017, 2018, and 2019 were averaged, thus avoiding the problem of fixed and random effects.

- Third, we built a database with these 41 variables for 28 facilities. The descriptive statistics of all the variables collected in the study are available in Table A1 in Appendix A.

### 3.2. Data Analysis

These data were processed using the statistical regression method, which is suitable for predicting a dependent variable using one or more independent variables [39]. The objective here was to develop an equation to predict one variable based on the knowledge of the other variables. Specifically, we used backward regression, which eliminates one by one the least significant variables until the remaining variables are all statistically significant (at $p < 0.05$). Thus, as in the study by Howat, Murray, and Crilley [19], in the final model, only the variables that significantly contributed to the prediction of aquatic centre attendance remained.



To do so, we carried out statistical processing (descriptive statistics and inference tests) to identify the explanatory variables and then develop a mathematical model to estimate aquatic centre attendance. MINITAB® 19.2020.1 software was used to perform these calculations:

- We first calculated the Spearman correlation coefficients between variables two by two, crossed with a test of significance ($p < 0.05$). As our data set did not follow a normal distribution (e.g., socio-demographic variables), the most appropriate correlation test was the Spearman rank order correlation coefficient [40]. We then removed six variables that were not significantly correlated with attendance: two variables related to the appeal of the proposed activity (category 1), two demographic variables (category 3), and two variables related to "public preferences" (category 4).
- Demographic data by subcategory (from 3.1. to 3.4.) were highly correlated. Consequently, to prevent multicollinearity, we retained the variable in each subcategory that was the most correlated with the variable to be explained, namely, attendance. Therefore, only four variables were linked to the catchment area demographics.
- In this stage, 17 variables potentially explaining attendance remained: ten from category 1 (appeal of the proposed activity), two from category 2 (economic variables), four from category 3 (demographic variables), and one variable from category 4 ("public preferences"). We then performed regressions on the best subsets, limited to 10 combined variables, to determine which regression showed the highest power to explain attendance.
- Finally, a succession of multiple backward regressions made it possible to remove the non-significant variables ($p < 0.05$) from those retained in this model.

Thus, we obtained the results presented below.

## 4. Results

The correlations of each of the variables with attendance are presented first; then, the predictive model is presented.

### 4.1. Spearman Correlations and Significance

The results are presented in Table 3. For each variable, we show the Spearman correlation coefficient with the variable to be explained, namely, attendance, and the *p*-value related to this correlation (We used Spearman coefficient to assess the correlation between each potential explanatory variable and the variable to be explained in pairs. This allowed us to eliminate from the outset those variables identified in the literature that had no impact on attendance in our sample). We thus present the following:

- The 13 remaining variables in categories 1, 2, and 4.
- The 6 variables excluded due to $p > 0.05$.
- The 4 variables retained in category 3 (characteristics of the catchment area; in bold in Table 3).

In Table 3, the 10 variables in italics and grey are those with the highest correlations with attendance.

The two variables most correlated with attendance were in category 1, which grouped together the variables that characterized the proposed activities in these centres. First, attendance was explained by the number of hours when the centre was open to the public (1.1; R = 0.781) and the number of hours when various activities were offered at the centre (1.2; R = 0.691). One variable dealt with facility size, the total area of all pools in m$^2$ (1.5.2; R = 0.601). Finally, in this first category, the communication budget appeared to be one of the variables strongly correlated with attendance (1.6; R = 0.679).

In the second category, one economic variable also appeared to strongly explain attendance. This was the normal entrance fee for adults (2.1; R = 0.633). Similarly, the number of public transport means (4.1; R = 0.628) was the only variable found to be strongly correlated with attendance in the fourth category ("public preferences").

**Table 3.** Correlations with the variable to be explained and significance of the correlation.

| Variable | R | *p*-Value | Variable | R | *p*-Value |
|---|---|---|---|---|---|
| *1.1.* | *0.781* | ≤0.001 | 3.1.1. | 0.560 | 0.002 |
| *1.2.* | *0.691* | ≤0.001 | 3.1.2. | 0.547 | 0.003 |
| 1.4.1. | 0.501 | 0.007 | 3.1.3. | 0.563 | 0.002 |
| 1.4.3. | 0.435 | 0.021 | **3.1.4.** | **0.569** | **0.002** |
| 1.4.4. | 0.578 | 0.001 | 3.1.5. | 0.549 | 0.002 |
| 1.4.5. | 0.572 | 0.001 | **3.2.1.** | **0.590** | **0.001** |
| 1.4.6. | 0.410 | 0.030 | 3.2.2. | 0.575 | 0.001 |
| 1.5.1. | 0.506 | 0.006 | 3.2.3. | 0.563 | 0.002 |
| *1.5.2.* | *0.601* | 0.001 | 3.2.4. | 0.575 | 0.001 |
| *1.6.* | *0.679* | ≤0.001 | 3.2.5. | 0.533 | 0.003 |
| *2.1.* | *0.633* | ≤0.001 | 3.2.6. | 0.456 | 0.015 |
| 2.2. | 0.495 | 0.007 | 3.2.7. | 0.450 | 0.016 |
| *4.1.* | *0.628* | ≤0.001 | 3.3.2. | 0.535 | 0.003 |
| | | | 3.3.3. | 0.552 | 0.002 |
| **Excluded Variable *p* > 0.05** | | | *3.3.4.* | *0.594* | 0.001 |
| 1.3. | −0.249 | 0.201 | 3.3.5. | 0.588 | 0.001 |
| 1.4.2. | 0.302 | 0.118 | 3.3.6. | 0.542 | 0.003 |
| 3.3.1. | −0.287 | 0.138 | 3.3.7. | 0.498 | 0.007 |
| 3.4.1. | −0.059 | 0.767 | *3.3.8* | *0.595* | **0.001** |
| 4.2.1. | 0.019 | 0.923 | *3.4.2.* | *0.632* | ≤**0.001** |
| 4.2.2. | −0.083 | 0.673 | *3.4.3.* | *0.604* | 0.001 |
| | | | 3.4.4. | 0.584 | 0.001 |

In bold: The 4 variables retained in category 3 (characteristics of the catchment area). In italics and grey: The 10 variables most correlated with attendance.

The variables characterising the catchment area were almost all significantly correlated with attendance, except for the variable quantifying the agricultural population of the territory (3.3.1), which is a population traditionally little represented in the categories of people practising physical activity. The second excluded variable referred to the socio-economic data related to the value that divides into two equal parts the standard of living of the households in the catchment area (3.4.1).

The last four variables were features of the catchment area (category 3). Thus, attendance depended on the number of inhabitants in the catchment area aged 15 years or over and working in mid-level professions (3.3.4; R = 0.594); the number of inhabitants in the catchment area aged 15 years out of work but not receiving unemployment benefits, such as students or housewives (3.3.8; R = 0.595); and the number of people employed in the catchment area (3.4.2; R = 0.632). We can assume that variables 3.3.4 and 3.4.2 were linked because they reflect the tertiarization of the French economy, which involves office jobs (mid-level professions) rather than blue-collar jobs. These job variables in the catchment area and mid-level professions were, therefore, linked. The last variable in category 3 was the number of unemployed in the catchment area (3.4.3; R = 0.604).

We also noted that the correlation between this last variable and the general adult admission fee was not significant (*p* = 0.142), which undoubtedly reflects specific pricing policies aimed at this population segment to facilitate their access to the centre. The pricing policy is generally decided by the centre owners, i.e., the public authorities [14].

Concerning the fourth category related to "public preferences", the two weather variables (4.2.1 and 4.2.2) did not appear to be significantly correlated with attendance, which can probably be explained by the nature of the variable itself. The retained average, which makes it possible to smooth out exceptional events, does not make it possible to consider specific climatic phenomena that would have an impact on attendance. In contrast, the number of means of public transport (with stops less than 300 metres from the centre) appeared to be a strong explanatory factor for attendance (4.1; R = 0.628). However,

these variables most strongly correlated with attendance did not necessarily constitute the combination of variables with the highest predictive power. Therefore, at this stage, we tested the 17 remaining potentially explanatory variables.

### 4.2. Development of an Explanatory Model of Aquatic Centre Attendance

Following the succession of backward regressions (As a reminder, stepwise regression is a statistical method that eliminates the potentially explanatory variables that are not correlated with the variable to be explained by a succession of tests. The final model includes all the explanatory variables directly correlated with the variable to be explained (in this case, attendance), with a significance threshold of $p < 0.05$ (i.e., a margin of error of 5%, commonly considered acceptable in the scientific community).) on the 28 cases, the model was ultimately made up of the below four variables to explain aquatic centre attendance. For each one, the regression with the variable to be explained was significant, with a $p$-value less than 5% ($p < 0.05$).

In Table 4, we note that the variance inflation factor (VIF) is between 1.15 and 1.60, which indicates the absence of multicollinearity in the final model according to the literature. Indeed, a VIF higher than 10 tends to indicate multicollinearity [41,42]; some authors suggest a threshold of 5 [43].

**Table 4.** Regression model.

| | Variable | Coeff | Coef ErT | Value of T | Value of $p$ | VIF |
|---|---|---|---|---|---|---|
| | Constant | −220,795 | 63,775 | −3.46 | 0.002 | |
| 1.4.3. | Total area of all the spaces for children's activities in m$^2$ | 58.1 | 19.8 | 2.94 | 0.007 | 1.15 |
| 2.1. | Normal entrance fee for adults | 35,302 | 13421 | 2.63 | 0.015 | 1.45 |
| 1.1. | Number of hours when the centre was open to the public | 39.7 | 11.3 | 3.52 | 0.002 | 1.60 |
| 3.3.8. | Out of work in 2013 | 0.703 | 0.205 | 3.43 | 0.002 | 1.16 |

The explanatory formula for aquatic centre attendance is the following:

$$\text{FREQ} = -220{,}795 + (58.1 \times \text{SUPENF}) + (35{,}302 \times \text{PXBASEMOY}) + (39.7 \times \text{NBHGPMOY}) + (0.703 \times \text{POPCS8})$$

The association of the four variables resulted in the coefficients of determination shown in Table 5.

**Table 5.** Coefficients of determination of the explanatory model of aquatic centre attendance.

| R$^2$ | 79.13% |
|---|---|
| Adjusted R$^2$ | 62.37% |

These results demonstrate that the proposed formula could predict more than 79% of the attendance observed in our sample of aquatic centres. The reliability of this formula for attendance prediction was nearly 63%.

To complement these results, we performed residual value analysis to evaluate the residuals of the model fits. This analysis is presented in Figure 1 below.

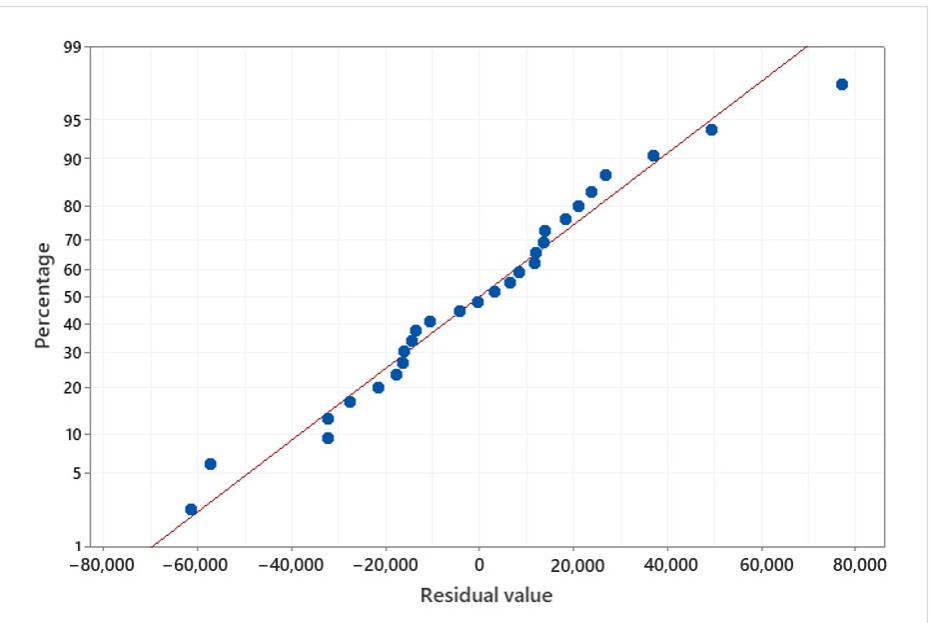

**Figure 1.** Distribution of the residual values of the model.

As we can see in Figure 1, the model fitted most observations. The few observations that the model did not fit precisely (e.g., residual value higher than +30,000 or lower than −30,000) concerned aquatic centres with low attendance compared with the average attendance observed in the sample (60,000 entries or less per year) or high attendance (200,000 entries or more per year). It is also possible to explain these inaccuracies by analysing the data of the four variables contributing to the model on these aquatic centres. Here, again, we find very different values in comparison with the average for the aquatic centres in the sample for two variables: number of hours when the centre was open to the public (1.1.) and number of people out of work in 2013 (3.3.8.). Thus, the inaccuracies of the model can be explained by the specific operation of the facility (e.g., longer-than-average opening times) or by local specificities (e.g., catchment area not adapted to the local context). As a reminder, for methodological reasons, we defined a uniform catchment area for all the aquatic centres in the sample, which ignores local specificities.

## 5. Discussion and Contribution

### 5.1. Discussion

In this study, we identified some of the variables that determine aquatic centre attendance to enrich our understanding of the factors that contribute to the attractiveness of these centres.

Hypothesis one was partially validated. For the variables in the first category, we showed that attendance was explained by the number of hours when activities were offered by the operator and by the number of hours when the centre was open to the public. These results confirm those of Howat, Murray, and Crilley [19]. They found that the variable best explaining attendance at Australian aquatic centres was the number of activities offered each week. Overall, the work related to the analysis of consumer experience quality highlights the importance of the quality of services provided in building consumer loyalty [24,26–28,33]. Our quantitative approach, therefore, proposes results that are consistent with those resulting from work based on qualitative variables. The surface area of all pools in m$^2$ was also highly correlated with attendance (1.5.2; R = 0.601). These results are consistent with the previous results related to the volume of activities offered, and the larger the spaces for activities are, the more activities can be offered. In fact, the number of hours when activities were offered was positively correlated both with the area of the pools (R = 0.522) and the area dedicated to fitness (R = 0.428). These

findings confirm the link established by Lau et al. [24] between the size of the facility and its attendance. Howat, Murray, and Crilley [19] indicated that the amount spent on communication by the facility was an explanatory variable for its attendance. Our results show a correlation of R = 0.679 between communication budget and attendance, which is the third highest correlation observed. Our results, therefore, confirm the results obtained by Howat, Murray, and Crilley [19] and show the importance of communication policy in an aquatic establishment in attracting the public.

However, in this category, the temperature of the water in the pool did not appear to be significantly correlated with attendance. However, in France, a publication in the Official Bulletin specifies the conditions for learning to swim by setting the ideal and minimum temperatures to be respected (Official Bulletin n° 32 of 9 September 2004). The temperature is, therefore, more or less the same in most swimming pools.

Regarding the economic variable that explained attendance, the general entrance fee for adults was positively correlated with attendance. This result is contrary to the logic of price elasticity in economics and was verified by Howat, Murray and Crilley [19] (R = −0.111, significant at $p < 0.01$). This result does not mean, however, that the more the fee increases, the more attendance increases; instead, it might indicate a willingness to pay more for broader services or those considered to be of better quality, and/or it may be linked to a catchment area more capable of paying a higher price. It should also be noted that the average admission fee in our sample was EUR 5.60, varying between EUR 3.9 and EUR 6.8, which is quite low. As a reminder, the pricing policy is generally decided by the public authorities that own the centres [14,44]. Consequently, our price elasticity shows that pricing is not exclusively based on the cost of the service, but also on political choices, such as social criteria [17]. For example, in France, user revenue only covers 22% of the operating costs of a public aquatic centre [45]. The economic variable representing direct competition for the aquatic centre appeared to be significantly correlated with attendance, but to a lesser extent (2.2; R = 0.495), which can probably be explained by the low density of facilities on the territory in France [46].

In total, 17 of the 19 variables characterising the catchment area were significantly correlated with attendance, including four variables at nearly 60%: two related to profession and socio-professional category (3.3.4; 3.3.8) and two related to the economic dynamics of the territory (3.4.2; 3.4.3). This result shows that it is imperative to consider the characteristics of the catchment area through a micro-localized study. These results are consistent with those reported by Moulard [47] and Scelles et al. [37] for French soccer stadiums. This relationship between attendance and the characteristics of the catchment area is an original result of this study. Indeed, earlier publications on aquatic facilities only considered the total population in the catchment area [19] and concluded that there was no link between population size and attendance. However, demographic data from INSEE allowed us to test 24 variables characterising the population of the catchment area according to socio-demographic and activity criteria. Generally, the demographic context between Australia and Europe, which is more densely populated, may explain the difference in the results of the catchment area. The size and demographic structure of the catchment area are based on a managerial approach aimed at identifying the impact of operating methods on quality, overall satisfaction, and loyalty [24,27–29]. Moreover, aquatic facilities in Europe were initially built according to demographic density criteria. It is, therefore, not surprising that this criterion does not appear to be as decisive as in other international studies. In our study, it was the characteristics of the population that determined the level of attendance at the facility.

Finally, the importance of the number of public transport lines serving the centre confirms the need to ensure public transport to access leisure facilities, particularly in peri-urban areas [48]. On the other hand, the weather criteria did not appear to be determining factors for attendance. Most of the facilities are indoors. Practice is, therefore, probably less dependent on external weather factors.

In addition to identifying the variables that contribute to explain attendance at an aquatic facility, our work proposes a predictive model of attendance at an aquatic centre

based on four variables. As a matter of principle, the variables selected by the model are discriminating in explaining variations in attendance from one facility to another. Thus, two of the four variables belong to category 1 ("proposed activity"), including variable 1.1 (the number of hours when the centre was open to the public), which was the variable most correlated with attendance, and a variable related to the children's aquatic space (1.4.3), which could influence the attendance of families. The economic variable retained by the model is the normal entrance fee for adults (2.1), which was the fourth most correlated variable with attendance. Finally, the last variable included in this model represents the population in the catchment area aged 15 years out of work but not receiving unemployment benefits, such as students or housewives (3.3.8), which was also highly correlated with attendance (the ninth most correlated variable). Three of these four variables are of the same nature as those identified by Howat, Murray, and Crilley [19] in the Australian case: fees per visit, programme opportunities per week, and catchment population. However, the fourth variable in their model was the promotion cost share, while we included a floor space variable that was not tested by the authors. This undoubtedly contributed to the fact that our model, with these four independent variables, predicted attendance at aquatic centres in France with an accuracy of over 79% (hypothesis 2 was validated), while the model proposed by Howat, Murray, and Crilley [19], who designed an explanatory model of attendance at Australian aquatic centres, showed an explanatory rate of 54.6%.

Even if these two models offer rather comparable results, care should be taken when directly comparing these results with studies with different research objectives studying very different local contexts.

*5.2. Contribution*

These results have of two major implications.

This study updates and expands on previous work carried out by Howat, Murray, and Crilley [19] on Australian aquatic facilities. It is, therefore, the first to investigate leisure sports facilities in Europe. The analysis was also thorough due to the number of potential explanatory variables that were tested. Particularly, 15 variables related to the characteristics of the facility (and its offer) and variables representative of the socio-demographic structure of the catchment area (n = 24), and not only the overall size of the population, were included. Finally, a predictive model of aquatic centre attendance was determined, with a power of over 79%. These elements make an innovative contribution to the research field focused on identifying factors that explain attendance at leisure sports facilities. The method used could be reproduced in other sports facilities and in other operating contexts (public, private, or public–private partnerships).

From a managerial perspective, this model constitutes a decision-making tool in two ways.

On the one hand, it allows a better assessment of attendance risk, and thus operating risk, of the facility to be performed. In addition, the highlighting of the importance of activity variables in attendance must be considered by the operators of these facilities. This should influence the definition of the offer by the operators and the marketing strategy that should be adopted towards the users–customers to optimise attendance. For a long time, local authorities have considered a facility to have costs of investment and operation. This work shows that they must also take into account the cost of animation in order to propose activities that generate attendance and thus have greater social utility [15,18,49]. A little-used facility ultimately costs slightly less to run (fewer reception, surveillance, and maintenance staff) but is less socially useful. Conversely, consideration should be given to the potential over-use of these facilities, which may have a negative impact on user satisfaction [50].

On the other hand, this model contributes to the reflection on the sizing of future facilities by encouraging the actors to take better account of the facility location, as informal interviews indicate that they tend to focus more on programmed projects that are based on

the facility characteristics rather than the environment. The programming of these facilities should also integrate the idea of modularity, which could, over time, allow facilities built for the long term to be adapted to the evolution of practices (for example, movable floors and walls, connectivity, etc.).

## 6. Conclusions

The operation of large sports facilities such as aquatic centres is a major challenge for public authorities, given their economic impact on both for the public and private sectors. Due to their size and complexity, these are projects that must be managed over the long term, and due to their nature, they inevitably raise diverse societal issues. Our approach seems essential, as we want to help the various actors to better size facilities and adapt them to the "micro-locality" [47]. Our model, although perfectible, thus offers a decision support tool for stakeholders in the business ecosystem concerned with constructing and operating aquatic centres. This highlights the importance of considering the specific characteristics of each centre's catchment area, for both programming decisions and evaluating the operational prospects. It also shows the need to ensure well-chosen activities to optimize attendance, thereby demonstrating the social utility of these facilities, which is a way to ensure financial investment in the public sector. This methodology can be applied to other types of sports–entertainment facilities, such as arenas, as well as fitness gyms and even indoor soccer facilities.

However, this work was based on the analysis of 28 facilities, i.e., the maximum number of facilities for which we could obtain 5 years of data from the largest French operator in order to have comparable management and operating methods. Indeed, the economic model of this operator is based on maximizing service, whereas others choose to minimize costs. Having 5 years of data made the smoothing of a possible exceptional year of operation possible. The generalisation of these results is, therefore, limited. Our model is only exploratory and should be tested on a larger number of facilities.

Finally, this work could be extended with a view to qualifying the effectiveness of the public service provided by these facilities. Indeed, today, it is no longer just a question of knowing how many people benefit from the facility but of creating indicators to evaluate the social impact of public policies according to the objectives pursued (health, education, sports, etc.). Existing official reports (such as those of the Court of Auditors in France) propose indicators, but they only focus on costs and are not related to social utility, nor do they include management data.

This opens two avenues. On one hand, this work could be extended within the framework of an approach aimed at analysing consumer experience quality [51] with the objective of supplementing the quantitative variables used in this research with qualitative variables characterising the management of these facilities in order to better assess the satisfaction of users of these facilities and the impact on attendance.

On the other hand, Liu, Taylor, and Shibli [14], and Ramchandani, Shibli, and Kung [18] used economic and management data to measure the operational effectiveness of public sports facilities (financing, use, customer satisfaction, etc.). The current difficulty lies in the access to and homogeneity of data on leisure sport facilities. Liu, Taylor, and Shibli [35], and Kung and Taylor [52] also used social indicators to qualify the social origin of users (age, gender, ethnicity, education, disability, etc.).

It seems that these are relevant and fruitful directions that have been opened and deserve to be extended to identify the most relevant indicators of social utility, but also to objectify the usefulness (or lack of) and effectiveness (or lack of) of publicly funded services.

**Author Contributions:** Conceptualization, A.B., B.E. and N.D.-R.; methodology, A.B., B.E. and N.D.-R.; validation, A.B., B.E. and N.D.-R.; formal analysis, A.B., B.E. and N.D.-R.; investigation, A.B., B.E. and N.D.-R.; data curation, A.B., B.E. and N.D.-R.; writing—original draft preparation, A.B., B.E. and N.D.-R.; writing—review and editing, A.B., B.E. and N.D.-R. All authors have read and agreed to the published version of the manuscript.

**Funding:** This research received no external funding.

**Institutional Review Board Statement:** Not applicable.

**Informed Consent Statement:** Not applicable.

**Data Availability Statement:** Data is unavailable due to privacy.

**Conflicts of Interest:** The authors declare no conflict of interest.

## Appendix A

**Table A1.** Descriptive statistics of all variables collected in the study.

| | Mean | Standard Deviation | Minimum | Median | Maximum |
|---|---|---|---|---|---|
| Attendance | 128,790 | 66,269 | 32,508 | 114,383 | 266,686 |
| 1.1. | 3061 | 708 | 1825 | 3007 | 4370 |
| 1.2. | 1412 | 531 | 743 | 1390 | 2595 |
| 1.3. | 28.362 | 0.444 | 27.900 | 28.210 | 29.600 |
| 1.4.1. | 3648 | 2052 | 900 | 3370 | 11,000 |
| 1.4.2. | 249.9 | 157.0 | 0.0 | 231.5 | 805.0 |
| 1.4.3. | 129.9 | 341.8 | 0.0 | 45.5 | 1830.0 |
| 1.4.4. | 184.1 | 276.1 | 0.0 | 125.0 | 1500.0 |
| 1.4.5. | 207.4 | 204.7 | 0.0 | 200.0 | 660.0 |
| 1.4.6. | 128.7 | 117.7 | 0.0 | 105.0 | 460.0 |
| 1.5.1. | 5.536 | 2.117 | 3.000 | 5.000 | 14.000 |
| 1.5.2. | 684.8 | 271.0 | 321.0 | 665.3 | 1612.0 |
| 1.6. | 26,032 | 12,229 | 7175 | 22,750 | 60,653 |
| 2.1. | 5.600 | 0.570 | 3.900 | 5.680 | 6.800 |
| 2.2. | 2.393 | 1.750 | 0.000 | 2.500 | 5.000 |
| 3.1.1. | 235,059 | 229,411 | 21,981 | 138,202 | 818,837 |
| 3.1.2. | 103,893 | 105,707 | 10,500 | 56,808 | 386,755 |
| 3.1.3. | 113,195 | 109,856 | 10,429 | 67,076 | 396,329 |
| 3.1.4. | 122,367 | 120,502 | 11,552 | 71,138 | 428,048 |
| 3.1.5. | 191,898 | 189,670 | 18,976 | 109,993 | 683,411 |
| 3.2.1. | 44,355 | 42,787 | 2963 | 27,978 | 154,942 |
| 3.2.2. | 46,111 | 46,286 | 2247 | 23,669 | 171,576 |
| 3.2.3. | 47,195 | 47,432 | 3256 | 27,649 | 166,902 |
| 3.2.4. | 45,792 | 44,277 | 4740 | 28,028 | 158,536 |
| 3.2.5. | 32,762 | 32,716 | 5167 | 19,668 | 137,660 |
| 3.2.6. | 17,864 | 19,330 | 2987 | 9582 | 88,929 |
| 3.2.7. | 2263 | 2635 | 275 | 1255 | 12,633 |
| 3.3.1. | 707 | 570 | 28 | 694 | 2576 |
| 3.3.2. | 6025 | 6867 | 689 | 3439 | 32,249 |
| 3.3.3. | 20,874 | 28,301 | 482 | 9913 | 118,215 |
| 3.3.4. | 28,856 | 29,636 | 1565 | 17,339 | 115,928 |
| 3.3.5. | 32,289 | 31,558 | 2597 | 19,080 | 118,699 |

**Table A1.** *Cont.*

|  | Mean | Standard Deviation | Minimum | Median | Maximum |
|---|---|---|---|---|---|
| 3.3.6. | 23,122 | 20,860 | 2419 | 15,588 | 75,049 |
| 3.3.7. | 47,662 | 47,799 | 8539 | 28,148 | 205,036 |
| 3.3.8. | 32,388 | 33,128 | 2202 | 15,933 | 113,045 |
| 3.4.1. | 1,494,500 | 1,195,821 | 144,677 | 1,226,829 | 4,505,966 |
| 3.4.2. | 107,288 | 107,866 | 4949 | 71,723 | 395,233 |
| 3.4.3. | 15,173 | 15,110 | 894 | 8220 | 50,374 |
| 3.4.4. | 22,966 | 28,667 | 2041 | 12,009 | 124,004 |
| 4.1. | 1.286 | 1.329 | 0.000 | 1.000 | 4.000 |
| 4.2.1. | 78.77 | 9.78 | 56.30 | 77.39 | 119.26 |
| 4.2.2. | 110.72 | 15.03 | 58.00 | 110.90 | 143.60 |

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
