# Peer review of "Predictive Modelling of Sports Facility Use: A Model of Aquatic Centre Attendance"

_sustainability, doi:10.3390/su15054142_

Round 1

Reviewer 1 Report

Thanks for the opportunity to review this work.

Although the paper deals with an interesting in a rather structured and clear way, it does present some critical points. The work requires major revision for publication.

First, I think, the paper lacks a description of the leisure sport sector. For example, what is the weight of the sector in terms of employees, of added value in the French economy?

Secondly, in data analysis and results there aren't descriptive statistics on the variables (i.e. mean, variance, asymmetry, kurtosis) and the results of the residual analysis are not provided.  Furthermore, why was Spearman' s coefficient chosen among the correlation indices? The interpretation of the final model should be better explained. Few considerations are made between the explanatory variabiles and the aquatic centre attendance.

Finally, the discussion and the conclusions need to be strengthened, they are rather generic and have little relation to the results of the analysis.

Author Response

Please see the attachment (pages 2 to 6).

Reviewer 2 Report

Interesting, current and important topic.
High quality item.
In abstract, The authors describe in great detail the research methodology used. Here, the research problem, the methodology used (in general) and the results of the research (in general) should be described.
The authors precisely indicated the purpose of the study.
It should be indicated whether the number of examined objects (28) meets the criterion of the minimum sample size. This information determines the possibility of generalizing the obtained research results to the general population.
There are no research hypotheses. They should be added.
The hypotheses should be addressed in the discussion of the results.
In the titles of the tables, you should rather not enter information: authors' calculations.
Selection of the option: "Can be improved" results from the fact that everything in science can be improved.

Author Response

Please see the attachment (pages 7 to 9).

Reviewer 3 Report

Reading the article entrusted to me for review was a pleasure. Both from the point of view of the editorial ease of the authors and from the scientific point of view. As a sports management professional, I appreciate the authors' insight in the area of preparing the determinants of the study. A factor in plus of the article is also the very open presentation of the research process with a very good discussion of the different stages of the process. The only reservations may be the selective choice of example subjects discussed in the article. The fact that they come from different countries may also be of great importance if only in terms of cultural or atmospheric differences for those attending swimming facilities. In the bibliography, only less than 35% of the entries are from the last 5 years (2018 and newer)

Author Response

Please see the attachment (pages 10 to 12).

Round 2

Reviewer 1 Report

I appreciated the effort made by the authors to improve and clarify some passages of the paper.

Therefore, I suggest to accept the paper with "minor revision" after checking the expression "logistic regression" used in the abstract.

Author Response

Please see the attachment (page 2).
